# Prevalence and associated factors of premenstrual syndrome among women of the reproductive age group in Ethiopia: Systematic review and meta-analysis

**Teshome Gensa Geta**[1]*, **Gashaw Garedew Woldeamanuel**[1☉], **Tamirat Tesfaye Dassa**[2☉]

**1** Department of Biomedical Sciences, School of Medicine, College of Medicine and Health Sciences, Wolkite University, Wolkite, Ethiopia, **2** Department of Midwifery, College of Medicine and Health Sciences, Hawassa University, Hawassa, Ethiopia

☉ These authors contributed equally to this work.
\* teshgen2006@gmail.com

**Data Availability Statement:** All relevant data are within the manuscript and its Supporting information files.

## Abstract

### Introduction

Premenstrual syndrome is a clinical condition characterised by the cyclic occurrence of physical and emotional symptoms, which can interfere with normal activity. It significantly affects the health-related quality of life and can result in decreased work productivity. The prevalence of premenstrual syndrome varies widely in different countries and different regions of the same country. Thus, this study was aimed to estimate the pooled prevalence of premenstrual syndrome and its associated factors among women in Ethiopia.

### Materials and methods

Published studies searched from electronic databases such as PubMed/Medline, google scholars, HINARI, Science Direct, Cochrane Library, and EMBASE were used. All studies done among women of the reproductive age group in Ethiopia and reported in the English language were included. The current study was reported using Preferred Reporting Items for Systematic Reviews and Meta-Analyses (PRISMA) guidelines. Two authors extracted the data independently by using Microsoft excel extraction format and transported to STATA 14 software for analysis. $I^2$ test was used to assess heterogeneity between the studies. A random-effect model was computed to estimate the pooled prevalence and associated factors of premenstrual syndrome. The prevalence and odds ratio with 95% confidence interval (CI) were presented using a forest plot.

### Results

After careful screening of 33 studies, nine studies were included in our systematic review and meta-analysis. The pooled prevalence of premenstrual syndrome in Ethiopia was found to be 53% (95% CI: 40.64, 65.36). Subgroup analysis by university versus high school

**Funding:** The author(s) received no specific funding for this work.

**Competing interests:** The authors have declared that no competing interests exist.

showed a pooled prevalence of 53.87% (95% CI: 40.97, 67.60) and 56.19% (95% CI: 6.80, 105.58), respectively. The pooled odds ratio shows that age at menarche, menstrual pattern and hormonal contraceptive use had no statistically significant association with premenstrual syndrome.

## Conclusion

More than half of the women under reproductive age group were experiencing premenstrual syndrome in Ethiopia.

## Introduction

The American College of Obstetricians and Gynecologists (ACOG) defined premenstrual syndrome (PMS) as a clinical condition characterised by the cyclic occurrence of physical and emotional symptoms unrelated to any organic disease that appear during the five days before menses and ends at four days after onset of menses in three consecutive cycles with sufficient severity that interfere with normal activity [1].

Premenstrual syndrome is diagnosed based on timing of the symptoms. The diagnosis starts with women's experience of at least one of affective or somatic symptoms before 5 days of menses in three prior menstrual cycles which is relieved with menstruation. The affective symptoms are depression, angry outbursts, irritability, anxiety, confusion, social withdrawal, and somatic symptoms are breast tenderness, abdominal bloating, headache, swelling of extremities. Premenstrual syndrome can only be proven after exclusion of other diagnosis that may better explain the symptoms. Again the symptoms should be confirmed by two prospective cycles with impairment of some facet of women's life [1].

The prevalence of premenstrual syndrome varies in different countries. For instance, the prevalence of premenstrual syndrome was reported as 12.2% in France [2] and 98.2% in Iran [3]. The global prevalence of premenstrual syndrome is 47.8% (95% CI: 32.6–62.9) [4].

Although the exact cause of premenstrual syndrome is not known, it is believed to be triggered by hormonal changes ensuing after ovulation [5]. Progesterone is the main factor behind PMS symptoms. The metabolites of progesterone; allopregnanolone and pregnanolone are potent neuroactive steroids. These hormones are positive allosteric modulators of gamma-aminobutyric acid (GABA). Gamma-aminobutyric acid (GABA) is the main inhibitory neurotransmitter in the brain and it is important for regulating stress, anxiety, alertness, and seizure. The progesterone metabolites; allopregnanolone bind with GABA receptor in the brain. This binding changes the configuration of the receptor and decreases its sensitivity to GABA. This lowers serotonin level, which gives rise to symptoms of premenstrual syndrome [5].

Premenstrual syndrome is associated with different socio-demographic factors like age, marital status, and living region [6]. Premenstrual syndrome is also associated with stress due to heavy duties, coffee intake, age at menarche, long menstrual cycles, and being sexually active [7]. Parent's income and previous history of depression were also associated with premenstrual syndrome [8].

Premenstrual symptoms severely affects the health-related quality of life, increase health care utilization, and decrease work productivity [9, 10]. The decreased productivity at work and performance at school is associated with a lack of concentration, motivation, and poor involvement in collaborative work [11].

Despite the negative impact of PMS on the health-related quality of life, less attention has been given to it. Determining the pooled prevalence of premenstrual syndrome at a country level gives a better figure than discrete primary studies. Therefore, this systematic review and meta-analysis study was aimed to estimate the pooled prevalence of premenstrual syndrome and its associated factors in Ethiopia.

## Materials and methods

### Search strategy

Systematic review and meta-analysis were done by using Published studies. The articles were searched by electronic databases such as PubMed/Medline, google scholars, HINARI, Science Direct, Cochrane Library, and EMBASE.

The key terms and search strategies used for intensive search were (((associated factors [Title/Abstract] OR risk factors [Title/Abstract]) OR ((prevalence [Title/Abstract] OR magnitude [Title/Abstract]) OR "prevalence"[MeSH Terms])) AND (((premenstrual syndrome [Title/Abstract] OR premenstrual dysphoric disorder [Title/Abstract]) OR premenstrual tension [Title/Abstract]) OR "premenstrual syndrome"[MeSH Terms])) AND (Ethiopia [Title/Abstract] OR "Ethiopia"[MeSH Terms]).

The Preferred Reporting Items for Systematic Reviews and Meta-Analyses (PRISMA) guidelines were strictly followed in this study [12]. A protocol of the study has been registered by the International Prospective Register of Systematic Reviews(PROSPERO)(ID: CRD42020162498).

### Study selection and eligibility criteria

This systematic review and meta-analysis study incorporated all original research articles that reported the prevalence and associated factors of premenstrual syndrome among women of the reproductive age group in different parts of Ethiopia and published until December 25, 2019. All fully available studies written in the English language were included without restriction on their study design. Further tracing of studies was done by direct contact with the corresponding author of the existing article through email.

Before incorporating those studies into our meta-analysis, we reviewed the title and abstract of each study. After selecting relevant studies, the full text was reviewed. Articles with no variables of interest were excluded from our analysis.

### Quality assessment

To assess the data quality, a critical appraisal was done by using the Joanna Briggs Institute Meta-Analysis of Statistics Assessment and Review Instrument (JBI-MAStARI) [13]. The tool includes ten questions that address the methodology of the study. The criteria under this question include the representativeness of the sample used, the way of participants recruitment, adequateness of the sample size, the detail description of setting and subjects, use of standard measurement and its reliability, appropriate statistical analysis done with sufficient coverage of identified sample size and identification of subgroups. The components of quality assessment tools were clarified briefly by discussion among researchers. The quality assessment of all articles was done independently by two researchers. After taking the final score of the assessments from the two researchers, any disagreement was resolved by the negotiation of a third researcher. Studies with a score of greater than or equal to 6 out of 10 were considered as high quality and those studies with less than 6 out of 10 were considered as low quality. Those studies with high quality were included in the analysis.

### Outcome of interest

In this study, the main outcome of interest was the prevalence of premenstrual syndrome and its associated factors. The variables considered for this systematic review were age, alcohol intake, heavy non-academic duties, sexual activity, hormonal contraceptive use, and menstrual histories such as early menarche and menstrual pattern. The independent variables included in the meta-analysis were age at menarche, menstrual pattern, and hormonal contraceptive use.

### Operational definition

In this study, the following operational definitions were used. Age at menarche was divided in to early if $\leq 12$ and late if $> 12$ years [14]. Moreover, the menstrual pattern was divided into a regular; cycle with a regular monthly pattern, and an irregular pattern; cycle with fluctuating monthly pattern as reported by respondents [15].

### Data extraction

The two authors (TG and TT) independently extracted all necessary data by using Microsoft excel extraction format. The extracted parameters are the first author's name, year of publication, study design, sample size, prevalence, and tools used for the diagnosis of premenstrual syndrome. It also includes the number of participants with PMS and those with no PMS in association with the factors; age at menarche, menstrual pattern, and hormonal contraceptives. Then, the extracted data were checked again by the three authors, and disagreement was solved by tracing back to original articles.

### Data analysis

Data from Microsoft excel were exported to STATA software version 14 for analysis. The prevalence and standard error of each study were considered to calculate the pooled prevalence of premenstrual syndrome. The pooled prevalence was presented by the forest plot. To reduce the random error on the point estimate of individual studies, subgroup analysis was done based on the study setting (High School and University). The heterogeneity between studies was assessed by the $I^2$ statistical test and p-values less than 0.05 were used to declare it. Due to the presence of significant heterogeneity, the random-effects meta-analysis model was used. The possible source of heterogeneity was checked by meta-regression analysis by considering the publication year, sample size, and quality of the study. Egger's test was used to assess publication bias, and a p-value of less than 0.05 was used to declare its statistical significance. Odds ratio (OR) with 95% CI was also presented in the forest plot to show the associated factors of PMS.

## Results

### Study selection

During the search for published primary research articles, we found thirty-three studies. After duplications have been removed, twenty studies were left. These studies were further screened by a review of their title and abstract. Then, eight studies were removed. Among these, six studies have duplication of its content with different citations and publications of the same paper in different journals and the remaining two studies were conducted outside of Ethiopia. Moreover, the outcome of interest was not reported in the three studies and therefore, excluded from this study. Finally, nine studies were selected and included in this systemic review and meta-analysis (Fig 1).

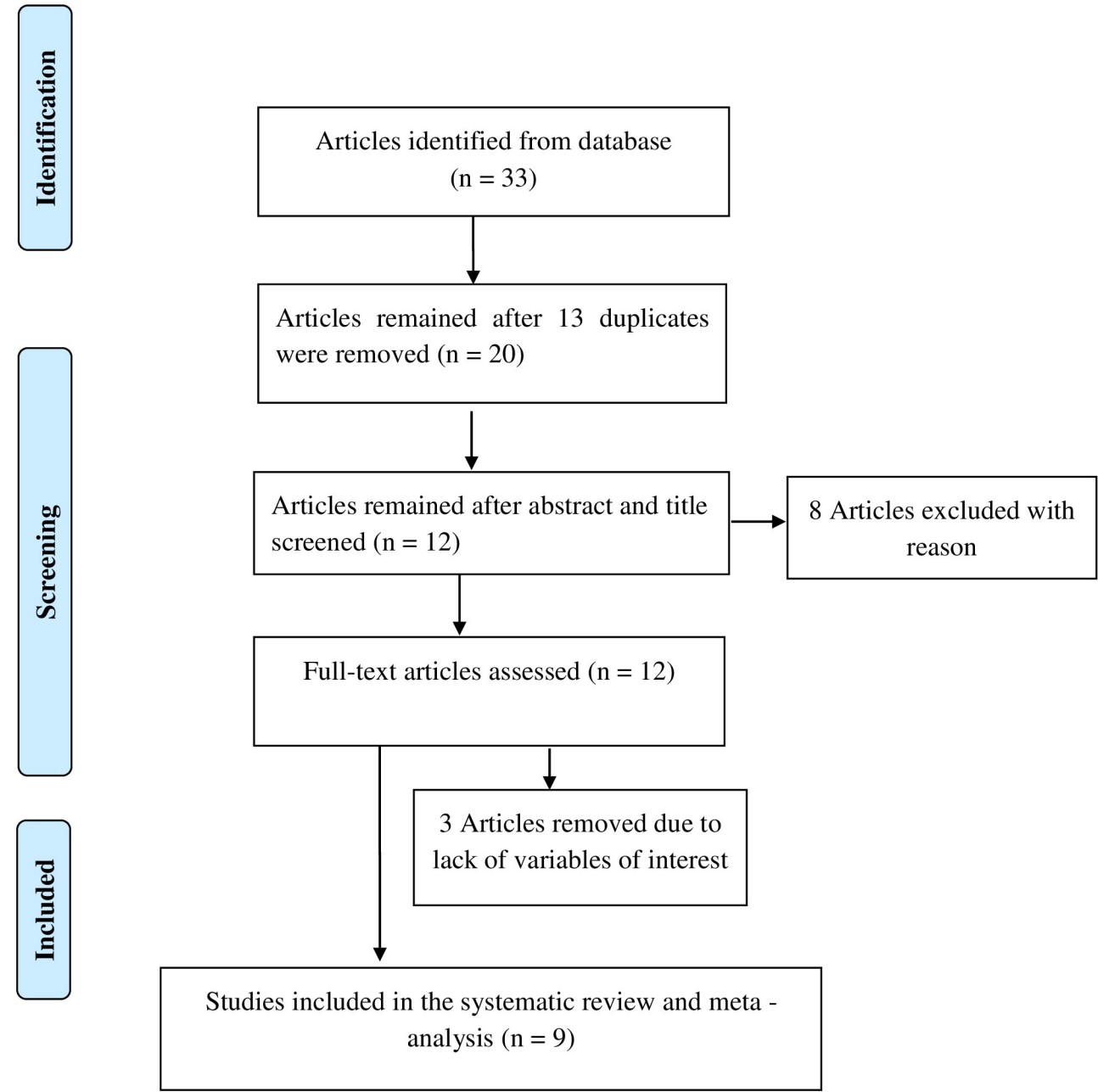

**Fig 1. Flow chart showing articles selection strategy for systematic and meta-analysis of prevalence and associated factors of premenstrual syndrome in Ethiopia, 2020.**

## Study characteristics

All studies included in this analysis were conducted using a cross-sectional study design [7, 8, 16–22]. For eight of these studies, the premenstrual syndrome scoring tool of Diagnostic and Statistical Manual of Mental Disorders, 4th Ed *(DMS IV)* was used while the American College of Obstetricians and Gynecologists (ACOG) was used for the remaining one study [7]. All the included studies were done from 2002 to 2019 and their sample size ranged from 181 [19] to 667 [17]. The total sample size considered for the pooled prevalence of premenstrual syndrome

**Table 1. Characteristics of studies included in the current systematic review and meta-analysis on the prevalence and associated factors of premenstrual syndrome in Ethiopia, 2020.**

| Author | Year | Study area | Study- design | Sample- size | Prevalence % | Study- setting | Tools |
|--------|------|-----------|---------------|--------------|--------------|----------------|-------|
| Abeje A. et al | 2015 | Debre Markos, Amhara region | Cross- sectional | 496 | 81.3 | High school | ACOG |
| Tenkir A. et al | 2002 | Jima,Oromia region | Cross-sectional | 242 | 27 | University | DSM -IV |
| Alemu M. et al | 2017 | Debre Berhan, Amhara region | Cross- sectional | 667 | 60.31 | University | DSM -IV |
| Asmare D. et al | 2013 | Debre Berhan, Amhara region | Cross- sectional | 321 | 41.12 | Community | DSM -IV |
| Desalegn J. et al | 2015 | Assosa, Benishangul Gumuz region | Cross-sectional | 519 | 58.6 | University | DSM -IV |
| Mossie T. et al | 2015 | Mekele, Tigrai region | Cross- sectional | 181 | 30.9 | High schools | DSM -IV |
| Muluken TS. et al | 2010 | Bahir Dar, Amhara region | Cross-sectional | 470 | 72.8 | University | DSM -IV |
| Tollosa & Bekele | 2013 | Mekele, Tigrai region | Cross- sectional | 223 | 37 | University | DSM -IV |
| Tsegaye D. et al | 2018 | Wollo, Amhara region | Cross-sectional | 254 | 66.9 | University | DSM -IV |

Note; ACOG: American College of Obstetrics and Gynecology, DSM-IV: Diagnostic and Statistical Manual of mental disorders IV.

in the current analysis was 3373. Moreover, all the studies were published articles, and their quality score range from 6 to 9 out of 10 points. Table 1.

## Prevalence of premenstrual syndrome

A wide range of prevalence rates was reported in the nine studies [7, 8, 16–22]. The lowest prevalence of PMS, 27.00% (95% CI: 21.41, 32.59) was reported in a study among Jimma University female students [16]. Whereas, the highest prevalence, 81.30% (95% CI: 77.87, 84.73) was reported in a study done among high school students of Debre Markos town [7]. From these nine studies, the pooled prevalence of premenstrual syndrome in Ethiopia was found to be 53% (95% CI: 40.64, 65.36). High heterogeneity was detected between the studies ($I^2$ = 98.4%, p < 0.001) (Fig 2).

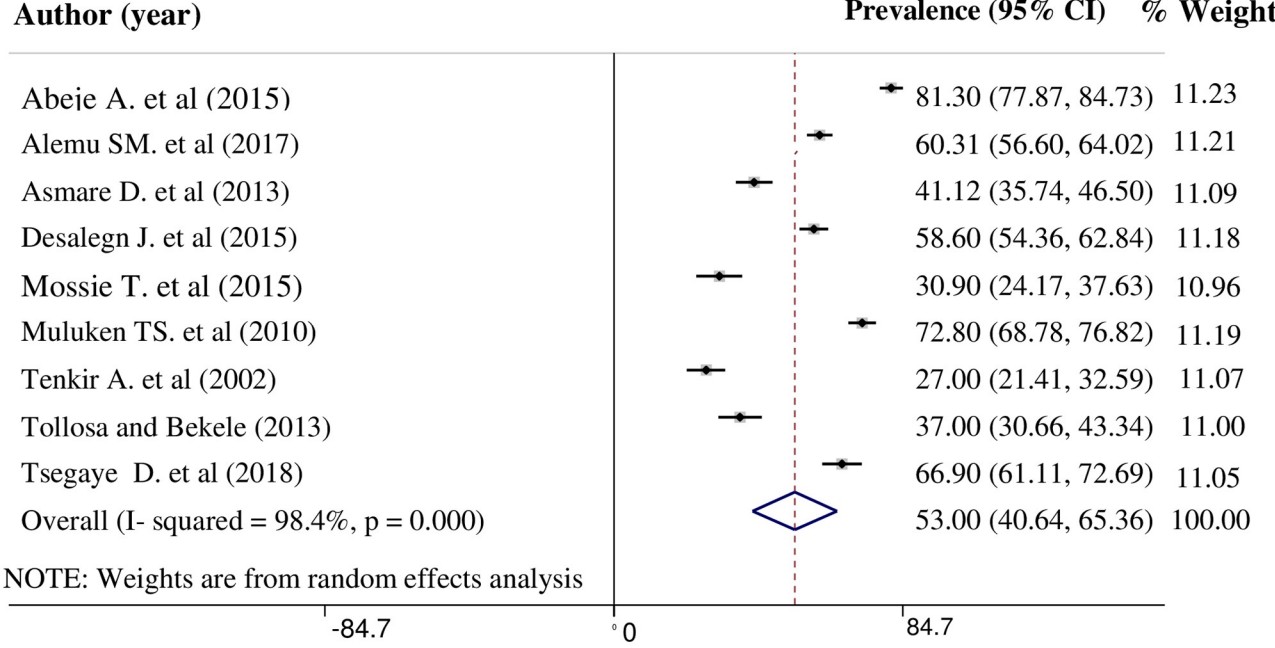

**Fig 2. Forest plot on the prevalence of premenstrual syndrome in Ethiopia.**

**Table 2. Heterogeneity related variables for the prevalence of PMS in the current meta-analysis (based on meta regression).**

| Variables | Coefficient | SE | P > | t | | (95% confidence interval) |
|---|---|---|---|---|
| Sample size | 0.028 | 5.896 | 0.996 | (-13.56895, 13.62482) |
| Year of publication | 0.025 | 5.896 | 0.999 | (-34.46264, 34.51393) |
| Quality of study | 0.027 | 5.896 | 0.976 | (-1.976467, 2.03019) |

**Note:** SE: Standard Error.

The statistical significance of factors considered to be a possible source of heterogeneity was investigated. Variables such as sample size, year of publication, and quality of the study were investigated using univariate meta-regression models. However, none of these variables was found to be statistically significant Table 2. Moreover, the Egger test showed statistically insignificant publication bias, P = 0.538.

## Subgroup analysis

To reduce the heterogeneity between the studies, subgroup analysis was done based on the study settings. Accordingly, the pooled prevalence of PMS among university students is 53.87% (95% CI: 40.97, 67.60). Although the heterogeneity of studies done at university showed improvement ($I^2$ = 97.8%, p < 0.001), significant heterogeneity still exists (Fig 3).

## Associated factors of premenstrual syndrome

Studies included under this systematic review and meta-analysis reported the factors associated with PMS. A study in high school in Debre Markos town showed that PMS has a statistically significant association with age, participation in heavy non-academic duties, coffee intake, early menarche, long menstrual cycles (> 35 days), and being sexually active [7]. The study at Jimma University also showed that age has a significant association with PMS [16]. Another study reported a previous history of depression and level of income has a significant association with PMS [8]. A study in Assosa Technical and Vocation College reported that there was a significant association between PMS and menstrual irregularity [AOR: 1.36, 95% CI (1.82, 2.25)]. The same study also showed that females not using contraceptive methods for the last six months were 1.92 times more likely to develop PMS as compared to their counterparts [18]. A study among Mekelle high school female students showed that income, cigarette smoking, alcohol, and contraceptive use had no significant association with PMS. However, PMS was significantly associated with duration of menses [AOR = 2.32, 95% CI (1.07, 5.05)] and early menarche [AOR = 3.11 95% CI (1.19, 8.12)] [19].

## Pooled odds ratio of associated factors

**Age at menarche.** Two studies witwere included in this analysis [7, 19]. Age at menarche has a significant association with PMS [7]. Contrariwise, another study found no significant association between age at menarche and PMS [19]. A meta-analysis of the pooled odds ratio shows that women with early menarche were 1.76 times more likely to develop PMS as compared to women with late menarche (OR: 1.76, 95% CI: 0.54, 5.70). However, the association was statistically insignificant. The heterogeneity test showed insignificant heterogeneity between the studies ($I^2$ = 69%, p = 0.073) (Fig 4).

**Menstrual pattern.** Three studies with a total sample size of 1545 were included to determine the association between menstrual patterns and PMS. Two of these studies show a significant association between menstrual patterns and PMS [18, 20]. However, the remaining study

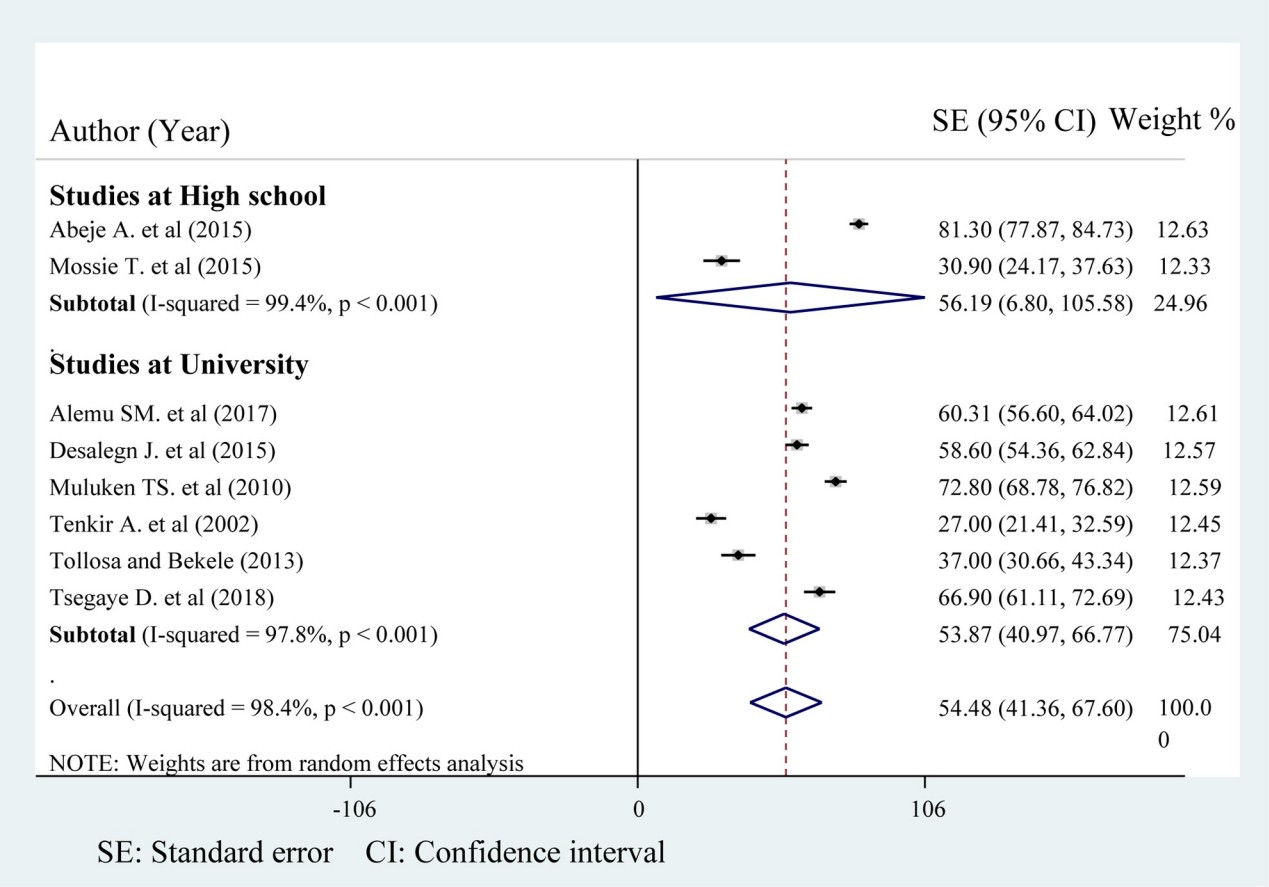

**Fig 3. Forest plot on the prevalence of premenstrual syndrome at different study setting in Ethiopia.**

shows a statistically insignificant association between PMS and menstrual patterns [7]. The pooled odds ratio also shows a statistically insignificant association between menstrual pattern and PMS (OR: 3.19, 95% CI: 0.94,10.80) (Fig 4).

**Hormonal contraceptive use.** Out of the two studies included in this current meta-analysis to determine an association between contraceptive use and PMS [18, 19], one study shows a significant association between hormonal contraceptive use and PMS [18], while the other one shows no significant association between use of oral contraceptive pills and PMS [19]. The pooled odds ratio shows that hormonal contraceptive use had no significant association with PMS (OR: 0.97, 95% CI: 0.23, 4.06) (Fig 4).

## Discussion

Regarding the prevalence and associated factors of PMS, this is the first systematic review and meta-analysis study in Ethiopia. Nine primary studies were included in the current systematic review and meta-analysis. This study assessed the prevalence and associated factors of PMS.

The pooled prevalence of PMS is 53% (95% CI: 40.64, 53.36). A subgroup analysis of the current study shows that the pooled prevalence of PMS among university students is 53.87% (95% CI: 40.97, 67.60). This result is in line with the other meta-analysis conducted worldwide [4]. It is also in line with other primary studies [9, 23]. Contrariwise, a higher pooled prevalence of PMS was reported in a meta-analysis study in Iran (70.8%) [24]. However, studies also

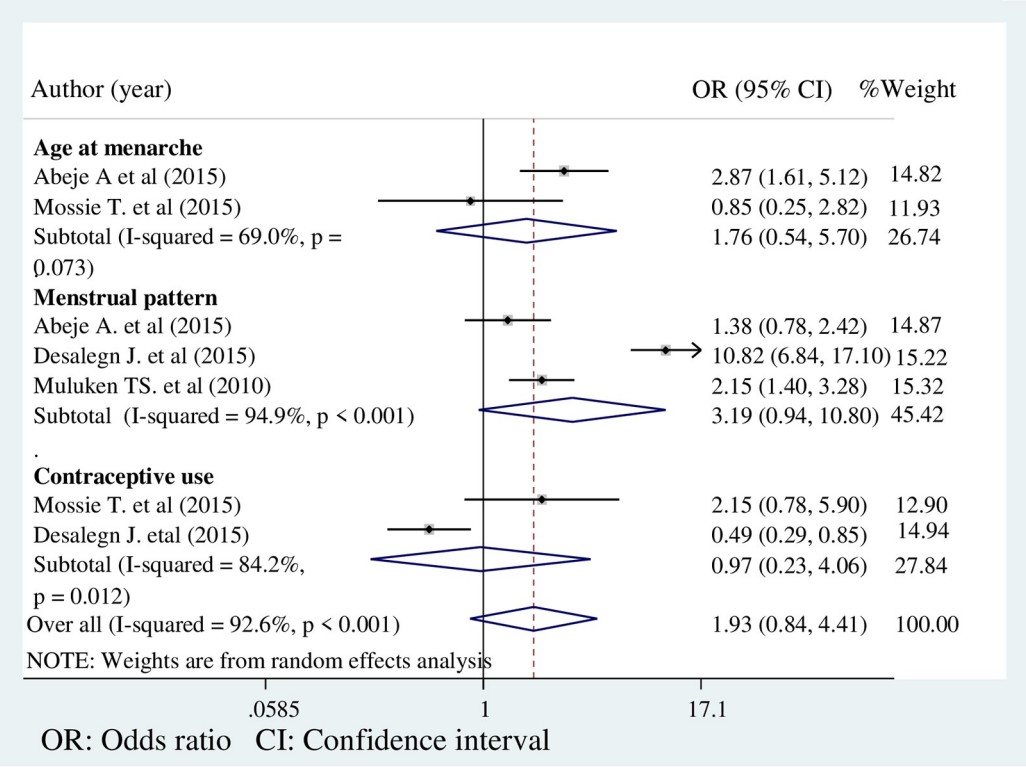

**Fig 4. Forest plot on the associated factors of premenstrual syndrome in Ethiopia.**

reported a lower prevalence of PMS as compared to the current study [2, 3, 25–27]. These variations from the current study might be due to differences in sociodemographic factors of the study participants and tools used for PMS screening. It is difficult to determine the true prevalence of PMS in different setup because of self-treatment, differences in availability and access to health services, definition, and diagnostic criteria, and cultural practices [4].

The other aim of the current study is to assess associated factors of premenstrual syndrome. Although PMS is a problem common to women, the clear cause is unknown. It probably has to do with hormonal changes during each menstrual cycle. It gets triggered and associated with different factors.

The pooled odds ratio of the current study shows that PMS has a statistically insignificant association with age at menarche. This result is in line with other studies [2, 9, 28, 29]. Contrariwise to the current study, other studies show that age at menarche has a significant association with PMS [30, 31]. This difference between studies may be due to sociodemographic differences between the study participants.

Among the nine studies included in the current analysis, two studies [20, 22] reported a significant association between menstrual patterns and PMS. This is in line with other studies [2, 26, 29]. But the pooled odds ratio shows an insignificant association between menstrual pattern and PMS (OR: 3.19, 95% CI: 0.94,10.80).

In agreement with another study [32], the pooled odds ratio of the current study shows that hormonal contraceptive use has an insignificant association with PMS. However, a population-based survey in the French found a significant association between PMS and hormonal contraceptive use [2]. This variation in the result may be due to a difference in types of hormones and time duration of hormone use among study participants.

The strength of this study lies in the fact that it is the first meta-analysis and systematic review of the subjects in Ethiopia. However, the study has the following limitations. The analysis was done with few studies, and these studies were limited to the Oromia, Amhara, Benishangul-Gumuz, and Tigrai regions of Ethiopia. Moreover, age at menarche was categorized into two as early ($\leq 12$) and late ($> 12$). This is dichotomous category is not sufficient to explain the association between age at menarche and PMS. Also, classifying menstrual pattern as regular and irregular based on women's report pose a recall bias.

## Conclusion

More than half of women under the reproductive age group in Ethiopia are experiencing premenstrual syndrome. The findings of this study provide valuable information for policymakers, health professionals, and other stakeholders to set appropriate implementation strategies to reduce the impact of PMS. Moreover, further large-scale primary studies that focus on a wide range of independent variables are needed.

## Supporting information

**S1 File. DOC declarations.**
(DOC)

**S1 Table. Data set on the prevalence of premenstrual syndrome in Ethiopia.**
(XLSX)

**S2 Table. Data set on associated factors of premenstrual syndrome in Ethiopia.**
(XLSX)

## Acknowledgments

We would like to thank Mr. Getachew Mulu Kassa for his support and guidance on the statistical work of this study.

## Author Contributions

**Conceptualization:** Teshome Gensa Geta.

**Data curation:** Teshome Gensa Geta, Gashaw Garedew Woldeamanuel, Tamirat Tesfaye Dassa.

**Formal analysis:** Teshome Gensa Geta.

**Investigation:** Teshome Gensa Geta, Tamirat Tesfaye Dassa.

**Methodology:** Teshome Gensa Geta, Gashaw Garedew Woldeamanuel, Tamirat Tesfaye Dassa.

**Project administration:** Teshome Gensa Geta.

**Resources:** Teshome Gensa Geta, Gashaw Garedew Woldeamanuel, Tamirat Tesfaye Dassa.

**Software:** Teshome Gensa Geta, Gashaw Garedew Woldeamanuel, Tamirat Tesfaye Dassa.

**Supervision:** Teshome Gensa Geta.

**Validation:** Teshome Gensa Geta.

**Visualization:** Teshome Gensa Geta.

**Writing – original draft:** Teshome Gensa Geta, Gashaw Garedew Woldeamanuel, Tamirat Tesfaye Dassa.

**Writing – review & editing:** Teshome Gensa Geta, Gashaw Garedew Woldeamanuel, Tamirat Tesfaye Dassa.

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
