## [Decision Letter · Decision Letter 0]

22 Jun 2020

PONE-D-20-03722

Prevalence and associated factors of premenstrual syndrome among women of reproductive age group in Ethiopia: systematic review and meta-analysis

PLOS ONE

Dear Dr. Geta,

Thank you for submitting your manuscript to PLOS ONE. After careful consideration, we feel that it has merit but does not fully meet PLOS ONE’s publication criteria as it currently stands. Therefore, we invite you to submit a revised version of the manuscript that addresses the points raised during the review process.

We look forward to receiving your revised manuscript.

Kind regards,

Nülüfer Erbil, Ph.D, Prof.

Academic Editor

PLOS ONE

Journal Requirements:

2.In your Data Availability statement, you have not specified where the minimal data set underlying the results described in your manuscript can be found. PLOS defines a study's minimal data set as the underlying data used to reach the conclusions drawn in the manuscript and any additional data required to replicate the reported study findings in their entirety. All PLOS journals require that the minimal data set be made fully available. For more information about our data policy, please see http://journals.plos.org/plosone/s/data-availability.

Reviewers' comments:

Reviewer's Responses to Questions

**Comments to the Author**

1. Is the manuscript technically sound, and do the data support the conclusions?

Reviewer #1: Yes

Reviewer #2: Partly

2. Has the statistical analysis been performed appropriately and rigorously? 

Reviewer #1: Yes

Reviewer #2: I Don't Know

3. Have the authors made all data underlying the findings in their manuscript fully available?

Reviewer #1: Yes

Reviewer #2: No

4. Is the manuscript presented in an intelligible fashion and written in standard English?

Reviewer #1: No

Reviewer #2: No

5. Review Comments to the Author

Reviewer #1: Title: Prevalence and associated factors of premenstrual syndrome among women of reproductive age group in Ethiopia: systematic review and meta-analysis

The study by Teshome Gensa Geta et al, on premenstrual syndrome is of importance as it affects women globally who are in their reproductive age. The quality of women’s life during PMS is greatly affected by cluster of emotional, physical, and behavioural symptoms. These symptom expressions can vary between a few days to couple of weeks. As PMS is associated with mood and behavioural symptoms, a study by Kimberly Ann Yonkers et al.,(Lancet. 2008 Apr 5; 371(9619): 1200–1210) clearly described its connection with the brain and further suggested two different methods to treat these symptoms.

In the present study, the authors have claimed to have conducted the first systematic review and meta-analysis study on the prevalence and associated factors of PMS in Ethiopia. The authors have pooled various published original research articles from Ethiopia (until December 25, 2019) on PMS and its associate factors as their dataset. The quality of the dataset had been assessed using the Joanna Briggs Institute Meta-Analysis of Statistics Assessment and Review Instrument (JBI-MAStARI) while data analysis had been performed using the STATA software version 14.

From the 33 original research articles, using various criteria the authors have chosen to use the data only from 9 studies which has the total sample size of 3373 and considered for further analysis. The authors show that the pooled prevalence of PMS in Ethiopia was 53%(95% CI: 4064, 65.36). Further, high heterogeneity had also been detected in their study samples (I2 = 98.4%, p < 0.001). The authors have taken various factors into consideration like sample size, year of publication and quality of study to investigate the possible source of heterogeneity using statistical methods, however, the results were found to be insignificant. Eager test too showed insignificant P-value of P = 0538.

In order to reduce the heterogeneity, the authors have also performed the subgroup analysis based on study settings and found that the pooled prevalence of PMS among university students was 53.87% (95% CI: 40.97, 67.60). The result shows that though significant heterogeneity had been observed, there was some improvement of heterogeneity at the university level.

The authors have also studied various associated factors of PMS and show that, age, irregular menses, duration of menses, menstrual pattern has a statistically significant association with PMS among some of the high school and university students. Further age at menarche also had a significant association with PMS. In contrast a study done at Mekelle high school female students showed that income, cigarette smoking, alcohol, and contraceptive use had no significant association with PMS.

Based on the analysis and the results, the authors conclude that pooled prevalence of PMS in Ethiopia was 53% indicating that half of the Ethiopian women were suffered from PMS in their reproductive years. This number may vary across countries due to various factors such as socio-demographic, individual life-style, marital status etc. The pooled odds ratio showed that age at menarche, menstrual pattern and hormonal contraceptive use had statistically insignificant association with PMS.

The limitation of this study was restricted only with high school and university students who live in Oromia, Amhara, Benishangul-Gumuz and Tigrai region of Ethiopia.

The authors have carried out a significant work in understanding the PMS in different regions of Ethiopia among high school and university students. I hope the authors may work with the larger data in the future which could also include working women, married/unmarried in their dataset. Further, as PMS symptoms are mostly associated with mood and behaviour, I would also suggest the authors to discuss and co-relate their findings with the receptors expressed in the brain, such as serotonin and its associated receptors.

PS: I have a small concern, the Figure 1 ‘Fig 1: flow chart showing articles selection strategy‘ have a very high similarity to the Figure 1 from the previously published article by Ashraf Direkvand-Moghadam et al., J Clin Diagn Res. 2014 Feb; 8(2): 106–109. I have provided the link to this article below:

https://www.ncbi.nlm.nih.gov/pmc/articles/PMC3972521/

Kindly make changes in your Figure 1 before it’s been accepted for publication. Further, I would also suggest the author to proof read the manuscript with a native English speaker to improve standard of English in the manuscript.

This manuscript may be accepted for publication in the current form after the suggested corrections as their objective of the had been addressed.

My best wishes to all the authors.

Thank you.

Reviewer #2: This analysis – designed to estimate the pooled prevalence of Premenstrual Syndrome, PMS, and its associated factors in Ethiopia – followed a reasonable strategy to review published studies. It concludes that that the majority of reproductive aged high school and college aged women do experience PMS. The paper needs to be strengthened to render that conclusion credible.

Essential ms changes that should be made before consideration for publication include

1. Offer the reader a clear definition of what constitutes PMS since no disease is associated per the definition offered. Well-known hormonal shifts have been clearly defined in the literature.

a. The current definition of PMS is too vague and could apply to any fertile aged prescient woman who undergoes the well characterized sequential peak in progesterone approximately 7 days after ovulation and its subsequent decline to undetectable levels by menses onset. Since estrogen levels also plunge as P levels do, the assertion that P declines are responsible for PMS should be explained or omitted.

b. How is the “syndrome” diagnosed? Self-administered questionnaire then scored using standard procedures?

i. Can the authors provide a supplement for interested readers to review?

c. Does a health care worker make an assessment after interview?

d. Does a woman have to complain of the problem for it to be identified as PMS?

e. Who makes the judgment that this is a “syndrome” rather than a normal variation in energy and mood cycles?

2. An English language editor is essential because the word usage is frequently erroneous. For example, these underlined words do not make sense and the reader is left guessing what the authors probably mean:

a. Line 57 each of the three precede menstrual cycles

b. Line 61 The… meta-analysis did worldwide

c. Line 69-70 with joining in heavy duties… initial menarche

d. Line 71 being sexually active among students

e. Line 73-4 may outcome

f. Line 78 has given on it.

g. Line 103 .for studies that were not shown the outcome of

h. Line 114 after a brief discussion on the tool.

i. Line 117 those studies with less than 5 scores

j. Line 195 A study at high school students

k. Line 212 In contrary to this study…

3. Outcome of Interest

a. Independent variables chosen should be improved

i. Age at menarche was simply divided into two groups: those 12 or younger and those older. This seems too gross a division to hope to find an effect of age on PMS if avg age of menarche in Ethiopia mirrors the US at about 12 yrs. It would be more meaningful to divide groups into at least 3 or better yet 5 age groups: very early, early, average, mildly late, and very late. If you did, it would be interesting to learn about the relationship between the syndrome and the age

ii. Menstrual pattern as regular or irregular should be clearly explained: did women keep records of their menses that scholars extracted data from? Recall has been shown to be not useful in multiple studies that showed strong relationships between menstrual pattern and hormonal outcomes.

4. Results

a. Figures 2, 3 and 4 each need a legend to explain the scale on the x axis. For example, on Figure 4, I wonder how a prevalence could be less than zero. Or why -84.7 is even labelled?

b. The study characteristics Tools should be explained so that readers can understand how they measure their “findings.” (Please see point 1a above).

c. The prevalence outcome beginning on line 170 is clearly stated. Once the reader knows how the measures were obtained, the prevalence [in more than half the women] can be better understood to evaluate the importance of the findings.

d. Beginning line 227 It would be interesting to learn what kind of contraceptives were used in the study that did and the study that did not find a relationship to PMS. Some drugs flatten out the hormonal environment; others elevate and depress hormonal levels in sequence.

5. The strength of the study statement is good. As well as the limitations. Except for the misuse of English in line 273: were suffered from PMS.

6. PLOS authors have the option to publish the peer review history of their article (what does this mean?). If published, this will include your full peer review and any attached files.

Reviewer #1: No

Reviewer #2: No

---

## [Author Response · Author response to Decision Letter 0]

1 Aug 2020

Response to comments from editor and reviewer

Manuscript ID number: PONE-D-20-03722

Prevalence and associated factors of premenstrual syndrome among women of reproductive age group in Ethiopia: systematic review and meta-analysis 

We would like to express our heartfelt gratitude to editor and reviewers for constructive comments and guidance which are extremely helpful to improve this manuscript. Here are point-by-point responses for the comment raised. We had thoroughly revised the manuscript and provided the amendments that have been made to the manuscript text. 

Response to Editor comments 

1. The manuscript thoroughly edited based on PLOS ONE's manuscript writing format.

2. The data from which result and conclusion drawn was attached as one of supporting information.

3. The captions for Supporting Information files were written at the end of manuscript 

Response to Reviewer 1 Comments

1. Similarity of Figure 1 on flow chart showing articles selection strategy with previous published work was resolved.

2. English language edition was done by language editor. 

 John Odjenimah, email: ighoyiwi25@gmail.com

Response to Reviewer 2 Comments

1. A. The definition of premenstrual syndrome was clarified. 

B – E. Comment on this sub – section concerning diagnosis of PMS is very important but the scope of the current study not extends up to the level of evaluating diagnostic tools. As describe in the manuscript, all primary studies included in the current study were cross sectional studies. Those studies done by using two standard tools to diagnosis PMS such as American College of Obstetrics and Gynecology (ACOG) and Diagnostic and Statistical Manual of mental disorders IV (DSM-IV). 

For clarification on diagnosis of PMS by using ACOG; diagnostic criteria were described in paragraph two in introduction section.

2. All the indicated (subsection A -K ) and other section of manuscript were edited for English language by language editor. 

3. Independent variables

I. Age at menarche was divided into two groups: those 12 or younger and those older. This is classification done by primary studies. Indeed, it has limitation to describe association with PMS. Thus, we described this as limitation of study on manuscript.

II. Dividing menstrual pattern as regular or irregular was based on women’s report which is not based on record during each menses. All the study available on this title were cross sectional. So, we indicated recall bias as one of limitation of study. 

4. A. The figures were edited accordingly and possible amendments were done. Figure 2 is forest plot that shows prevalence of PMS in Ethiopia. The labeled 84.7 is maximum confidence interval level for prevalence of PMS. Figure 3 also show prevalence of PMS in different settings. In both forest plots, dotted line is pooled prevalence and forests shows how weighted prevalence from each study spread around pooled prevalence. 

Figure 4 shows odds ratio of associated factors. It may be mis-communication, there is no prevalence indicated as less than zero in manuscript.

B. Diagnostic criteria of PMS was described in paragraph two of introduction section. 

C. This comment was applied in the different sections of manuscript

 5. Language edition to whole part of manuscript was done

---

## [Decision Letter · Decision Letter 1]

2 Sep 2020

PONE-D-20-03722R1

Prevalence and associated factors of premenstrual syndrome among women of reproductive age group in Ethiopia: systematic review and meta-analysis

PLOS ONE

Dear Dr. Geta,

Thank you for submitting your manuscript to PLOS ONE. After careful consideration, we feel that it has merit but does not fully meet PLOS ONE’s publication criteria as it currently stands. Therefore, we invite you to submit a revised version of the manuscript that addresses the points raised during the review process.

We look forward to receiving your revised manuscript.

Kind regards,

Nülüfer Erbil, Ph.D, Prof.

Academic Editor

PLOS ONE

Reviewers' comments:

Reviewer's Responses to Questions

**Comments to the Author**

1. If the authors have adequately addressed your comments raised in a previous round of review and you feel that this manuscript is now acceptable for publication, you may indicate that here to bypass the “Comments to the Author” section, enter your conflict of interest statement in the “Confidential to Editor” section, and submit your "Accept" recommendation.

Reviewer #1: All comments have been addressed

Reviewer #2: (No Response)

2. Is the manuscript technically sound, and do the data support the conclusions?

Reviewer #1: Yes

Reviewer #2: Yes

3. Has the statistical analysis been performed appropriately and rigorously? 

Reviewer #1: Yes

Reviewer #2: Yes

4. Have the authors made all data underlying the findings in their manuscript fully available?

Reviewer #1: Yes

Reviewer #2: Yes

5. Is the manuscript presented in an intelligible fashion and written in standard English?

Reviewer #1: Yes

Reviewer #2: Yes

6. Review Comments to the Author

Reviewer #1: My best wishes to the authors and I wish the authors may carry out the study with a larger sample size across all over Ethiopia in the future.

Reviewer #2: The authors did a fine job of revising the manuscript and I have only a few remaining comments. I will not need to review the paper again and am fine with publishing it once these are considered.

1. 2 typos need to be fixed: first on line 60: ‘relieved’ is the correct spelling; not relived; then on line 64 add an “s” to pluralize the word “cycle”

2. The quality assessment paragraph beginning with line 115 is nice but there seems to be no mention of the relative quality assessments of the 9 studies that are analyzed. The careful reader will wonder which of the 9 were high-quality papers and whether these produced a more consistent outcome than putting them all in one basket. I assumed there was no quality restriction for inclusion. But I am left uncertain. The results did not provide this information. Or I missed it?

3. Line 295, I suggest “suffering” is not an appropriate term. We really cannot tell whether cyclic variation in mood, and energy which alters ones “normal activity” causes pain and suffering. In fact, the energy swing itself might be part of the human condition of what constitutes normal. Women are not robots and cyclic variation in energy and mood sufficient to alter how one engages in the work and other activities is a rational response to hormonal and metabolic change. I would suggest using the word “experiencing” rather than “suffering”. Considering that about half of women experience these swings might suggest that medicalizing such experience is not such a wise idea.

7. PLOS authors have the option to publish the peer review history of their article (what does this mean?). If published, this will include your full peer review and any attached files.

Reviewer #1: No

Reviewer #2: **Yes: **Winnifred Cutler

---

## [Author Response · Author response to Decision Letter 1]

14 Sep 2020

Response to reviewer 1 comment

 Constructive comment for future progress in the area was well accepted 

Response to Reviewer 2 comments

1. Two words with type error have been corrected

Line 60: relived is replaced by relieved

Line 64: cycle is replaced by cycles’

2. Clarification on quality assessment and selection of articles has been done by adding the following sentence. 

Line 127 in quality assessment under method’s sections: Those studies with high quality (quality score of 6 and more) were included in the analysis

Line 178 in study characteristics under result section: All the studies were published articles and their quality score range from 6 to 9 out of 10 points.

3. Line 295, comment on the word usage was considered and the appropriate term was replaced 

Line 296: suffering is replaced by experiencing

---

## [Editor Report · Decision Letter 2]

20 Oct 2020

Prevalence and associated factors of premenstrual syndrome among women of reproductive age group in Ethiopia: systematic review and meta-analysis

PONE-D-20-03722R2

Dear Dr. Geta,

We’re pleased to inform you that your manuscript has been judged scientifically suitable for publication and will be formally accepted for publication once it meets all outstanding technical requirements.

Kind regards,

Nülüfer Erbil, Ph.D, Prof.

Academic Editor

PLOS ONE
---

## [Editor Report · Acceptance letter]

26 Oct 2020

PONE-D-20-03722R2 

Prevalence and associated factors of premenstrual syndrome among women of the reproductive age group in Ethiopia: systematic review and meta-analysis 

Dear Dr. Geta:

I'm pleased to inform you that your manuscript has been deemed suitable for publication in PLOS ONE. Congratulations! Your manuscript is now with our production department. 

Kind regards, 

on behalf of

Dr. Nülüfer Erbil 

Academic Editor

PLOS ONE